# Early Olfactory Environment Influences Antennal Sensitivity and Choice of the Host-Plant Complex in a Parasitoid Wasp

**DOI:** 10.3390/insects10050127

**Published:** 2019-05-03

**Authors:** Martin Luquet, Olympe Tritto, Anne-Marie Cortesero, Bruno Jaloux, Sylvia Anton

**Affiliations:** 1UMR IGEPP, Agrocampus Ouest, 49054 Angers, France; martin.luquet@agrocampus-ouest.fr (M.L.); tritto.olympe@gmail.com (O.T.); bruno.jaloux@agrocampus-ouest.fr (B.J.); 2UMR IGEPP, Université Rennes 1, 35000 Rennes, France; anne-marie.cortesero@univ-rennes1.fr

**Keywords:** *Aphidius ervi*, Braconidae, early experience, electroantennography, host choice, olfactometry

## Abstract

Early experience of olfactory stimuli associated with their host–plant complex (HPC) is an important driver of parasitoid foraging choices, notably leading to host fidelity. Mechanisms involved, such as peripheral or central modulation, and the impact of a complex olfactory environment are unknown. Using olfactometer assays, we compared HPC preference of *Aphidius ervi* Haliday (Hymenoptera:Braconidae) females originating from two different HPCs, either with the other HPC in close vicinity (complex environment) or without (simple environment). We also investigated antennal responses to volatiles differentially emitted by the two respective HPCs. In a simple environment, HPC of origin had an influence on olfactory choice, but the preferences observed were asymmetric according to parasitoid origin. Electroantennographic recordings revealed significant sensitivity differences for some of the tested individual volatiles, which are emitted differentially by the two HPCs. Besides, presence of an alternative HPC during early stages modified subsequent parasitoid preferences. We discuss how increased olfactory complexity could influence parasitoid host foraging and biological control in diversified cropping systems.

## 1. Introduction

In order to find suitable hosts, female parasitoids of phytophagous insects use a wide array of sensory cues, generally emitted directly by the hosts themselves or by the plants on which they feed [1,2]. Whereas specialist species generally rely on innate preferences [3], such innate preferences for the most suitable host in generalist species are in many cases modified by experience. The influence of experience allows this to take host availability into account [4,5]. Learning of host-related cues during larval development or just after adult emergence in generalist species can thus lead to host fidelity, i.e., the preference for the host–plant complex on which they developed [6]. This mechanism could help parasitoids to effectively forage for the most abundant species when resource availability varies over generations [7], but may also represent a first step towards speciation [8,9]. For many parasitoid species olfaction has been identified as crucial for host detection and indeed, female parasitoids have been shown to be able to learn odors associated with their hosts, either before or after emergence from the host [3]. So far, little is known about which level in the olfactory pathway is modified upon learning of host cues in parasitoids. Modulation of the olfactory system upon associative learning in social insects seems to essentially be located within the central nervous system [10], but in a few cases modification of odor detection has been found in different insects [11,12]. Antennal odor detection has been investigated in several parasitoid species, but whether odor detection is modified by experience before or during emergence is not known. 

In the parasitoid *Aphidius ervi* Haliday (Hymenoptera:Braconidae), a generalist species which can develop on different host species without host-associated genetic differentiation [13], several studies have shown that females are able to learn odors from their host–plant complex (HPC) during pre-imaginal stages, both before and just upon emergence [14,15]. This mechanism leads to host fidelity in this species: When given a choice between cereal–*Sitobion avenae* (Fabricius) and legume–*Acyrthosiphon pisum* (Harris) complexes, an *A. ervi* female is more attracted by the HPC on which it developed [16]. This induction of host preference coincides with transcriptomic differences between females reared on different HPCs: Genes involved in neuronal growth and development, signaling pathways and olfactory detection are differentially expressed according to the HPC of origin [17]. Whether the differential antennal gene expression also leads to differences in antennal HPC volatile detection has not been investigated so far.

All studies revealing hostfidelity have been performed with insects developing on uniform simple HPCs, although in a natural environment, different hosts and their associated plants, emitting various volatile blends, might be present in close vicinity. Whereas several studies indicate that pest insects are less efficient in localizing their host plants in a complex olfactory environment, especially if repellent plants are associated with suitable host plants [18,19], it is still debated if this complex olfactory environment also affects natural enemies such as parasitoids [20,21]. An influence of surrounding plant volatiles on host choice has been indicated in *Aphidius rhopalosiphi* females: They seem not only to learn the odor of the HPC they emerge from (*Metopolophium dirhodum* on wheat), but also the combination of HPC volatiles with volatiles of plants present in the vicinity (tomato) [22]. However, it is still unknown if parasitoid females change their host plant preference when they develop in an environment where several of its potential hosts are present. 

In this study we investigated how early experience with HPC volatiles influences host preferences in *A. ervi*. We first tested the hypothesis that pre-imaginal learning of HPC odors leads to host fidelity in parasitoids reared in simple environments, with either one of two HPCs (*A. pisum* on faba bean and *S. avenae* on barley) in our populations using four-way olfactometers. Via electroantennographic recordings we examined if early experience might modify antennal detection of individual volatiles emitted by the two HPCs. We then tested the hypothesis that the presence of an alternative HPC close to the developing parasitoid alters its preferences as an adult.

## 2. Materials and Methods 

### 2.1. Insects and Plants

Two populations of *Aphidius ervi* parasitoids were collected as mummies from two aphid species representing major hosts: *A. pisum* (collected in pure faba bean fields in Seiche sur Loire (France) in spring 2016) and *S. avenae* (collected in pure wheat fields in Saint Sulpice sur Loire (France) in June 2016). In the laboratory, *A. pisum* was reared on faba bean (*Vicia faba* L.) and *S. avenae* on barley (*Hordeum vulgare* L.), forming two HPCs, ApF and SaB respectively. Before the experiments, both populations were reared on their respective HPC for over 50 generations. Plants, aphids and parasitoids were reared under long-day conditions (16L:8D) at 22 °C (day) and 20 °C (night). Humidity conditions were 80% relative humidity (day) and 90% relative humidity (night) for aphids and 70% relative humidity for parasitoids.

#### Parasitoid Conditioning

Insects from both laboratory populations (*ApF*-origin and *SaB*-origin) were divided into three rearing conditions:-Simple environment, without host switching: Parasitoids from each population had been raised on their respective HPC for over 50 generations and were reared in two separate climatic chambers for at least two generations. Thus, females used for experiments were exposed only to the odor cues of their natal HPC during their larval and pupal development.-Simple environment, with host switching: Females originating from each population reared in a simple environment for many generations were then allowed to oviposit on the alternative HPC, in the corresponding chamber. Females hatching from the resulting mummies were used for experiments. Thus, experimental insects originated from one HPC strain, but in the last generation developed on the alternative HPC (and were exposed only to the odor cues of this HPC).-Complex environment: Parasitoids from each population were raised on their respective HPC in a mixed environment, i.e., they were raised on one HPC in the presence of the other HPC. Cages containing the two HPCs were placed next to each other in the same climatic chamber, resulting in a distance of only a few centimeters between mummies on their HPC and the alternative HPC and thus odor exposure to both HPCs. Females were used for experiments after at least 2 generations in their complex environment.

For each parasitoid population in each rearing condition, mummies were collected daily in rearing cages and transferred in a separate cage without aphids, so that emerging females had no oviposition experience. Each day, newly emerged females were then placed with males in another separate cage containing an un-infested plant corresponding to their HPC of origin. After 24 to 48 hours, these females were considered mated and were used for behavioral or electrophysiological experiments.

### 2.2. Behavioral Experiments

Individual females from both origins, ApF and SaB, and each rearing condition, simple environment without host switching, simple environment with host switching and complex environment, were given the choice between both HPC odors in four-way olfactometers [23,24], placed in a white wooden case illuminated from above by two fluorescent white 18 W cold light tubes. Experiments were done at 22.5 ± 0.9 °C, 38% ± 4% relative humidity and a pressure of 1006 ±6 hPa. For stimulation, we used barley and faba bean plants of the same age, infested with 50 corresponding aphids for 48 to 72 h (stimulation HPC), placed in a glass vessel. Two outlets of each glass vessel were connected with silicon tubing to diagonal inlets of the four-way olfactometer. Stimulation HPCs were changed every day. A constant airflow of 0.3 L min^−1^ was produced by a sucking PTFE (polytetrafluoroethylene) membrane pump (KNF lab, France), connected to the central opening of the olfactometer and resulting in an air flow of 75 mL min^−1^ in each branch. The air entering the glass vessels was cleaned by an active charcoal filter and humidified by passing through a bottle filled with distilled water. A filter paper square was placed on the floor of the olfactometer and limits between the four odorant branches were drawn on it to follow parasitoid position. Because preliminary experiments had shown that parasitoid movements were very difficult to follow when they approached the center of the olfactometer, they circled too rapidly to track them reliably, we delimited a central zone with undetermined choice. The time spent in this area was not considered as a choice for either HPC (Figure 1). This central zone was designed as narrow as possible, but allowing a realistic decision on whether a parasitoid position corresponds to a choice or not. For each assay, a single parasitoid was introduced on the filter paper through the central hole of the olfactometer and its movements inside the olfactometer were recorded during 600 s, using the sequenceR interface implemented in R software [25]. Olfactometers were cleaned with detergent after each individual assay. Total times spent in each branch (HPC 1.1 + HPC 1.2, HPC 2.1 + HPC 2.2) and in the central zone were calculated. Time spent in both branches with the same stimulus was summed, and those sums were compared to evaluate relative HPC attractiveness. For each group of parasitoids (according to population origin and rearing conditions), 28 to 33 individual females were tested in the four-way olfactometer and each group of parasitoids was tested on at least three different experimental days.

### 2.3. Electroantennography 

In order to evaluate if antennal sensitivity to host odors depended on the HPC of origin of females, we evaluated their electroantennographic responses to volatiles that were differentially emitted by either HPC. For each HPC, we chose two commercially available volatiles that a) were not emitted in detectable amounts by the un-infested host plant, b) were emitted by the aphid–plant complex, c) were either not emitted, or emitted at negligible amounts by the alternative complex after aphid infestation [26,27,28,29,30]. For the *A.pisum*/faba bean complex, we selected octanal and α-pinene, two host-induced plant volatiles emitted by that HPC that are also individually attractive to *A. ervi* [26], and for the *S. avenae*/barley complex, sulcatone and sulcatol that are emitted after aphid infestation in different cereal–aphid complexes and can in certain cases be individually attractive to *Aphidus* parasitoids [27,28,30]. However, the behavioral attractiveness of the chosen individual volatiles is not necessarily relevant for the investigation of differences in antennal sensitivity.

For stimulation, volatiles were diluted in decadic steps in hexane and amounts of 10 ng to 100 μg in 10 μL of hexane of each volatile were applied onto a piece of filter paper inserted into a Pasteur pipette. A stimulation pipette with 10 μL of hexane on a filter paper was used as a control. To establish dose-response relationships, the five doses were tested in an ascending order and a control stimulation was applied before and after each dose-response series.

For electroantennographic (EAG) recordings, parasitoids of both populations under the first rearing conditions (simple conditioning environment, without host switching) were used. Mated 2 to 3-day-old females originating from one of the two HPC were decapitated. A glass electrode filled with Beadle-Ephrussi Ringer [31], connected to ground with a silver wire, was introduced into the cut end of the head. The tip of one antenna was then cut with fine scissors and the cut end was introduced into a second glass electrode. The electrodes were connected to an amplifier (axoclamp 2B, Molecular Devices) and the obtained signals were digitalized using an IDAC-4 device and registered using EAG Pro software (Syntech, Kirchzarten, Germany). The antenna was super-fused by a constant air stream (0.3 m/s) and odor pulses of 300 ms through the stimulating pipettes were introduced into the constant airflow using a stimulation device (Stimulus controller CS 55, Syntech, Kirchzarten, Germany). To correct for decline in EAG amplitudes over time due to antennal fatigue, EAG amplitudes were normalized in relation to the averaged responses to hexane applied before and after stimulation of the five doses of a tested volatile compound, following this formula: $$Normalized\;EAG\;response{\;}_{\left(volatile\right)}=\;\frac{2\;\times \;Abs.\;response{\;}_{\left(volatile\right)}}{Abs.\;response{\;}_{\left(Hexane\;1\right)\;}+\;Abs.\;response{\;}_{\left(Hexane\;2\right)}}$$
Where Abs.response = absolute EAG response, Hexane 1 = hexane control before dose-response, and Hexane 2 = hexane control after dose-response.

### 2.4. Statistical Analysis

For behavioral experiments, Friedman tests were used to compare the time spent in each zone of the olfactometer (ApF HPC, SaB HPC, central zone), followed by Wilcoxon tests for multiple comparisons. Even though our main interest was to compare the time spent in each stimulus zone (ApF HPC and SaB HPC), we included the time spent in the central zone in our analysis, because individuals spent largely varying times in this zone and excluding it would have biased our analyses. All statistical tests were performed using R software (R Core Team 2018).

EAG responses to each molecule were analyzed separately using linear mixed models (LMM) using the lme4 package [32]. Standardized antennal responses were modelled according to the HPC of origin of parasitoids and stimulus amount. An interaction term was added to consider that species responses could differ only at certain doses. Individual identity was included as a random factor, nested within insect origin, to account for individual variation. Dependent variables were log-transformed to linearize dose-response relationships and fulfil the assumptions of the models. After checking the validity of the models and when main effects were found significant, origin-dependent response variation at each dose was evaluated using pairwise comparisons of least-square means (lsmeans package [33]). 

## 3. Results

### 3.1. Influence of Early Experience in a Simple Environment

#### 3.1.1. Influence of Early Experience on HPC Choice

In both experiments with females originating from a simple environment (with and without host switching), *A. ervi* females spent far less time in the central zone (“undetermined choice”) than in the branches containing odor stimuli (Figure 2, Table 1). Independent of their conditioning, *A. ervi* females never spent significantly more time in the branches with odors from the SaB HPC than in the branches with odors from the ApF HPC (Figure 2).

Insects reared in simple environments exhibited an asymmetric response, according to their HPC of origin. Females originating from the ApF HPC, without host switching, spent significantly more time in the olfactometer branches with odors from their HPC of origin than in the branches with the alternative HPC odors (SaB) (Figure 2A, Table 1). We obtained the same result when females were reared on the SaB HPC and switched for one generation on the ApF HPC (Figure 2C, Table 1). On the other hand, females originating from the SaB HPC spent the same amount of time in the branches with odors of either HPC, whether they had been reared long-term on SaB HPC, or reared on the ApF HPC and switched for one generation on the SaB HPC (Figure 2B,D, Table 1). 

#### 3.1.2. Influence of Early Experience on Antennal Sensitivity

Significant dose-dependent EAG responses were obtained for all four tested compounds in females reared on both HPCs (*p* < 0.001 for all volatiles). There was no significant effect of HPC origin on averaged EAG amplitudes for three of the four tested volatiles: Sulcatol (χ^2^ = 1.6, *p* = 0.21), octanal (χ^2^ = 0.8, *p* = 0.37) and α-pinene (χ^2^ = 1.8, *p* = 0.17) (Figure 3). For sulcatone, however, antennal responses in the population originating from the ApF HPC were significantly weaker than in the population originating from the SaB HPC (χ^2^ = 10.2, *p* = 0.001) (Figure 3A). In addition, the interaction between HPC origin and dose was significant for sulcatol (χ^2^ = 10.2, *p* = 0.037). For the dose 10 μg, antennal responses in the population originating from the ApF HPC were significantly weaker than in the population originating from the SaB HPC (*p* = 0.047) (Figure 3B). Another significant interaction effect was discovered for α-pinene (χ^2^ = 13.1, *p* = 0.001). The response amplitude at the lowest tested dose (0.01 µg) was significantly stronger in the population originating from the ApF HPC, than in the population originating from the SaB HPC (*p* = 0.005) (Figure 3C). Note, however, that the dose-response relationship for α-pinene was much less pronounced than for the other tested compounds and response amplitudes were generally small (Figure 3C). 

### 3.2. Influence of Early Experience in a Complex Environment

Compared to insects reared in a simple olfactory environment, parasitoids lost the asymmetrical response when they had been exposed to a complex environment during early stages. Females spent significantly more time in the branches with odors from the ApF HPC than in the branches with odors from the SaB HPC independently of the HPC they were reared on (Figure 4A,B, Table 1).

## 4. Discussion

### 4.1. Adaptive Implications of Experience-Modulated Host Choice

A preference for the volatiles emitted by the *A. pisum*/faba bean HPC was shown in populations of *A. ervi* collected on this complex and reared on it for many generations, but also for populations collected and reared on the alternative HPC but switched on *A. pisum*/faba bean for one generation only. This result confirms that early learning is involved in olfactory HPC choice, rather than a genetic determinism due to host adaptation [13,16,34]. Parasitoid responses were asymmetrical, as rearing *A. ervi* on the *S. avenae*/barley HPC did not lead to a preference for this HPC. This result contrasts with a previous work by Daza-Bustamante et al. [16]: In this study, when given the choice between *A. pisum*/alfalfa HPC and *S. avenae*/wheat HPC, females from both origins preferred their respective HPC. This might be either due to the different host plants used, a different bioassay (wind tunnel), or to genetic differences between French and Chilean populations. Indeed, in Chile *A. ervi* has been introduced very recently [35] and is subject to a strong bottleneck effect [36]. 

For many parasitoids, preference modulation through early experience of host-related cues is a key mechanism leading to host fidelity: Females developing on a host will be more likely to be attracted and to oviposit on their natal host compared to an alternative host [37]. Consistent with our results, it was indeed shown that *A. ervi* reared on *A. pisum* prefers to attack its natal host [6]. The main hypothesis to explain host fidelity is that it may allow parasitoids to adjust their preference to the most common host in their nearby environment, while allowing changing this preference over generations [6,7]. However, considering this hypothesis, we would also have expected preference for the natal host to occur for females developing on *S. avenae*. The fact that the learning process is not symmetrical in our populations may be due to differences of resource quality between aphids. Indeed, the “mother knows best” hypothesis states that when given the choice between different hosts, parasitoid females should choose the one that will maximize fitness of progeny [38,39]. Earlier studies showed variable results about suitability of both hosts for *A. ervi*, either regarding Chilean [13,40], or European strains [41,42]. However, several of these studies indicate that *A. pisum* might indeed be a better host than *S. avenae* for *A. ervi* [40,42]. This could explain why the *S. avenae*/barley complex was not found more attractive than the alternative HPC in any of our tested situations. The validity of the “mother knows best” hypothesis is, however, debated as mothers do not always choose the host that will confer the highest fitness to their descendants [39,43]. What our experiments suggest is that both host suitability and probability of finding a host may influence parasitoid female preferences, leading to asymmetrical patterns. Females born on *A. pisum* would have a high chance of finding a high-quality host in their nearby environment, leading to an olfactory preference for *A. pisum*. When born on *S. avenae*, females would have a lower chance of finding a high-quality host and a higher chance to find a lower quality host, leading to equal attractiveness of cues associated to either host. The results observed here, may thus reflect a trade-off between host quality and availability.

### 4.2. The Effect of Experience of Antennal Sensitivity to HPC-Emitted Volatiles

EAG recordings of responses to volatiles that are differentially emitted by the two HPCs revealed a significant shift in the dose response curve to sulcatone, a compound typically emitted by cereal plants infested with cereal aphids [27,28,30]: Antennae of *A. ervi* originating from the ApF HPC exhibited a lower sensitivity for this compound than antennae of *A. ervi* originating from the SaB HPC. For the second compound, characteristic for the same HPC, sulcatol, significant differences in response amplitudes were only found for the second highest dose tested. For α-pinene, a significantly higher antennal sensitivity was found for the lowest tested dose in females originating from the ApF HPC, but this difference needs to be interpreted with caution, because responses to this compound were generally low and there was no clear dose-response relationship in the EAG amplitudes. No significant difference was revealed in the EAG responses to octanal between females of the two HPCs. Our physiological data revealed at least some differences between females with experience on the different HPCs, and agree with previous data, which showed a differential expression of genes involved in odor perception in *A. ervi* according to its HPC of origin [17]. The asymmetric changes in antennal sensitivity are at least partially correlated with asymmetric behavioral differences between insects originating from the two HPCs. However, this type of correlation between behavioral preferences and antennal sensitivity is not automatic. Differential antennal sensitivities in the two populations might lead as well to an increase as to a decrease in attractiveness. In addition, it remains to be investigated whether the small differences in peripheral odor detection are at all sufficient to explain the behavioral differences between *A. ervi* females raised on the two different HPCs. To interpret our physiological data, several points have to be taken into consideration. EAG recordings, reflecting the sum of receptor potentials across the entire antenna, do not provide a high resolution to detect small differences in sensitivity. Experience-dependent modulation of antennal odor sensitivity, correlated with a behavioral response increase, has previously been shown in a male moth, along with expression changes of antennal olfactory genes, but in this case, recordings from individual pheromone-sensitive receptor neurons were performed [12]. Nevertheless, it is more common that odor experience modifies sensitivity in the central rather than in the peripheral olfactory system in insects [44]. Furthermore, we only tested a few selected molecules so far, which have been described to be differentially emitted by the two HPCs investigated [26,28], but HPCs emit a vast number of additional molecules and central processing of complex odor blends might also be important for decision making in addition to detection of individual compounds at the peripheral level.

### 4.3. Influence of Environment Complexity 

This study, to our knowledge, is the first report of a change in HPC preference due to the perception of volatiles from a neighboring HPC by parasitoid females during early life. Previous studies in another parasitoid species had shown that volatiles from alien plants could be learned as part of the bouquet of an HPC: Females responded specifically to the blend they had experienced earlier [22]. Here, however, we show that females experiencing odors from another HPC can change their preference to this HPC as compared to the HPC they developed on, whereas they do not exhibit a preference when their experience is restricted to their own HPC. This result highlights the fact that ecological conclusions derived from host fidelity experiments must be taken with caution. Indeed, in natural settings, parasitoids often face a complex olfactory environment [45]. Detection of a wide variety of olfactory stimuli during parasitoid development could interact with host-associated odor learning and outcomes for subsequent parasitoid foraging behavior are difficult to predict. On one hand, learning alien cues could generate “noise” during the pre-imaginal learning process and disturb future host or mate selection for the adult parasitoid. On the other hand, there are situations in which learning stimuli associated with an alternative host could be adaptive, because it would provide the parasitoid with more complete information of the resources present in its environment. In our case, *A. ervi* raised on the SaB complex changed its preference to the ApF HPC when it was present in the nearby environment. If indeed *A. pisum* is a better host for *A. ervi*, our results would then support the “mother knows best” hypothesis. Identifying the presence of *A. pisum* during early stages (even if it is not the natal host) would help the future *A. ervi* mother making more adaptive foraging decisions, i.e., ovipositing on the host that will maximize progeny. The consequences of learning alien cues during developmental stages on foraging behavior and fitness remain to be investigated, however. In any case, these results suggest that the consequences of early learning not only depend on the host–plant complex but also on its environment, notably if alternative host species are present. Early learning is often presented as the initial mechanism of host speciation [9]. The modulation of host fidelity via integration of the odor of other HPCs available in the environment described in this study could counteract or at least weaken a host speciation process. The upholding of a generalist life style could be adaptive, reducing the evolutionary costs of host specificity, when the preferred host is rare in the environment [13]. This might also have implications for biological control, particularly in diversified systems. Considering our example again, intercropping systems mixing legume crops with cereals allow *A. pisum* and *S. avenae* to coexist in the same agroecosystem [46]. The presence of both HPCs could reduce host fidelity and make *A. ervi* females more prone to switch from one HPC to another. As these two aphids partially overlap phenologically in the field, this mechanism could lead to a better exploitation of both host populations and a better aphid regulation.

## 5. Conclusions

Behavioral experiments with *A. ervi* females from our laboratory-reared populations originating from two HPCs confirmed that early experience may modulate odor preference in this species. However, we revealed an asymmetrical response to the HPCs tested. Only females reared on the ApF HPC preferred their HPC of origin, whereas females reared on the SaB HPC did not exhibit any preference. This asymmetric response was at least partially correlated with differences in the antennal sensitivity to individual volatiles that are differentially emitted by the two HPC: i.e., Antennae of *A. ervi* females reared on the ApF HPC were less sensitive to individual volatiles emitted by the SaB complex. We thus revealed a modulation of antennal sensitivity through HPC experience. More importantly, our experiments revealed that early learning by *A. ervi* could be influenced by the complexity of the olfactory environment as females reared on the SaB HPC acquired a preference for the ApF HPC when reared in an environment with both HPCs. This could have consequences for biological control in diversified agroecosystems such as legume-cereal intercrops. 

## Figures and Tables

**Figure 1 insects-10-00127-f001:**
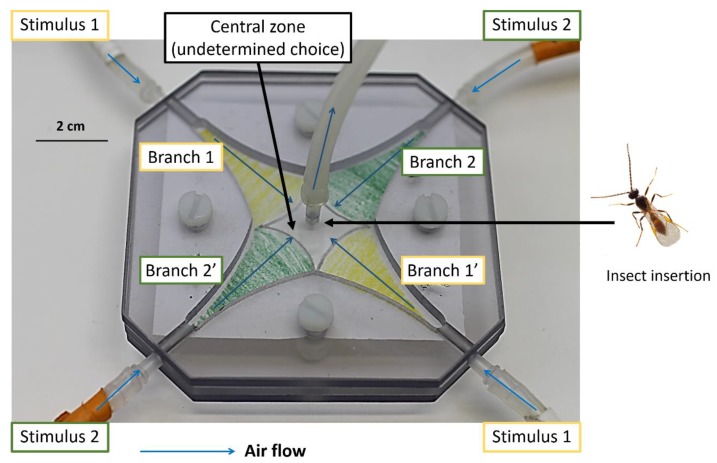
Four-way olfactometer setup used for the behavioral assays. A constant air flow allows diffusion of olfactory stimuli in each branch through polytetrafluoroethylene (PTFE) tubes connected to vessels with infested plants. The same colors, added for illustration only, indicate branches containing odors from the same host–plant complex (HPC). The diamond-shaped “central zone” in the center of the olfactometer is considered as zone of undetermined choice. After allowing the odors to diffuse in the device, parasitoids were inserted through a hole in the center of the olfactometer and observed during 600 s.

**Figure 2 insects-10-00127-f002:**
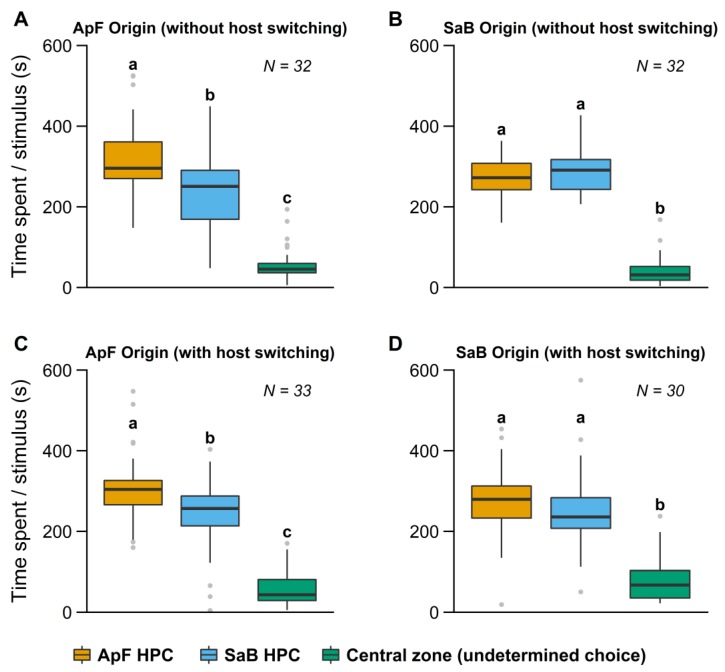
Time spent in the different branches of the olfactometer by *Aphidius ervi* females reared in simple environments. Females were reared on their respective HPC (*Acyrthosiphon pisum*/faba bean = ApF or *Sitobion avenae*/barley = SaB) (**A**,**B**) for several generations, (**C**,**D**) for one generation (host switch). Boxplot boundaries indicate the first and third quartiles and black lines within plots indicate the median for each treatment. Whiskers length equal to 1.5× interquartile range, other points are outliers. Bars with the same lower-case letters indicate no significant differences (Wilcoxon test).

**Figure 3 insects-10-00127-f003:**
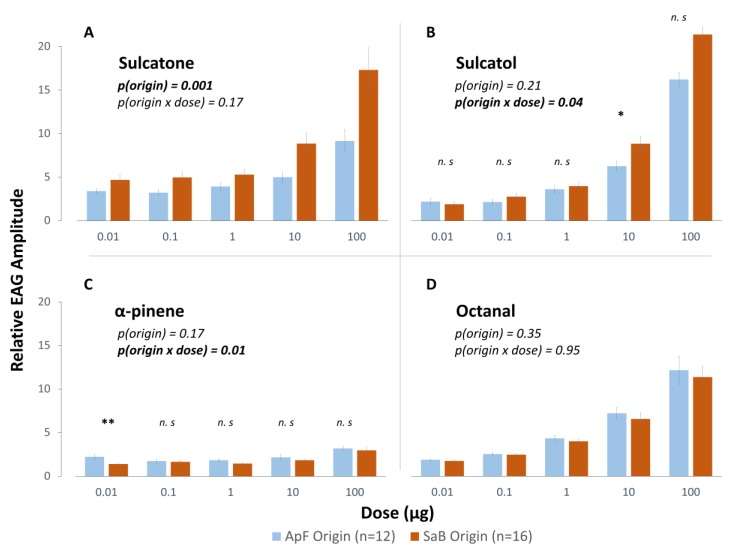
Mean (± SE) relative electroantennographic (EAG) amplitudes of *A. ervi* females reared in simple environments to volatiles emitted by the *S. avenae*/barley HPC (SaB): (**A**) Sulcatone and (**B**) Sulcatol; and to volatiles emitted by the *A. pisum*/faba bean HPC (ApF): (**C**) α-pinene and (**D**) octanal. Displayed *p*-values are associated to the “origin” term and to the interaction term between parasitoid origin and volatile dose in linear mixed models. Levels of significance for pairwise comparisons: n.s: Non-significant, * *p* < 0.05, ** *p* < 0.01.

**Figure 4 insects-10-00127-f004:**
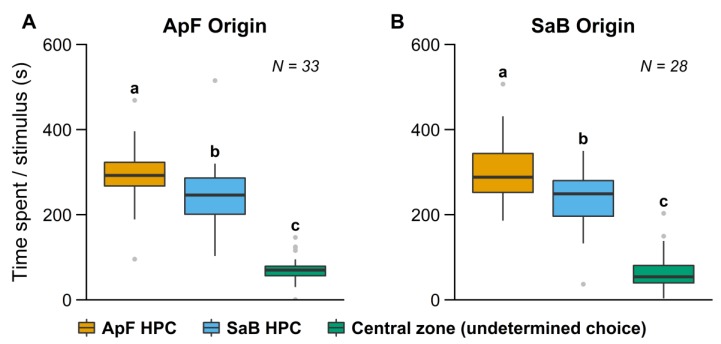
Time spent in the different branches of the olfactometer by *A. ervi* females reared in complex environments. Females were reared for several generations (**A**) on the *A. pisum*/faba bean HPC (ApF) or (**B**) on the *S. avenae*/barley HPC (SaB). Boxplot boundaries indicate the first and third quartiles and black lines within plots indicate the median for each treatment. Whiskers length equal to 1.5× interquartile range, other points are outliers. Bars with the same lower-case letters indicate no significant differences (Wilcoxon test).

**Table 1 insects-10-00127-t001:** Summary statistics for the behavioral experiments.

Environment	Treatment	Friedman’s Test (df = 2)		Pairwise Comparisons (*p*-value) ^1^		
		χ^2^	*p*	ApF vs. SaB	ApF vs. Central	SaB vs. Central
Simple	ApF (no host switching)	44	**<0.01**	**0.005**	**<0.01**	**<0.01**
	ApF (host switching)	46	**<0.01**	**0.034**	**<0.01**	**<0.01**
	SaB (no host switching)	48.3	**<0.01**	0.68	**<0.01**	**<0.01**
	SaB (host switching)	34.9	**<0.01**	0.22	**<0.01**	**<0.01**
Complex	ApF (no host switching)	48	**<0.01**	**0.011**	**<0.01**	**<0.01**
	SaB (no host switching)	36.5	**<0.01**	**0.048**	**<0.01**	**<0.01**

^1^*p*-values for pairwise comparisons were corrected with the false discovery rate method. SaB: *Sitobion avenae*/barley complex, ApF: *Acyrthosiphon pisum*/faba bean complex. Bold values indicate significant differences (*p* < 0.05).

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
