# Peer review of "Early Olfactory Environment Influences Antennal Sensitivity and Choice of the Host-Plant Complex in a Parasitoid Wasp"

_insects, 2019, doi:10.3390/insects10050127_

Round 1
Reviewer 1 Report
The paper is well written and the study well conducted however I recommend some changes on the content of the manuscript. In fact, I suggest to completely delete the study on EAG.
The overall EAG study (and the results and discussion on this) is confusing in this study since it does not bring a strong inputs in the overall manuscript. The authors used commercial compounds but not compounds or blend coming directly from their HPC under their experimental and rearing conditions. In addition, concluding that “A. ervi females reared on the ApF HPC were less sensitive to volatiles associated to the SaB complex.” is abusive since they used commercial compounds but not from compounds coming directly from their HPC in their both experimental and rearing conditions.
Moreover, concerning the EAG study, in Mat. & Met. section, the authors chose sulcatone and sulcatol for EAG done for the S. avenae/Barley complex based on the fact that these volatiles are emitted after aphid infestation in different cereal-aphid complexes BUT are they attractive to A. ervi as argued for the selection of the 2 volatiles used for the A.pisum/Faba bean complex? this aspect is not clear.
In this context, I suggest to the authors to delete the entire EAG study of their manuscript (somehow this is a too preliminary study).
I have also the minor points to address:
In the results section, in line 198 it might be Fig 2 and not only Fig. 2a? to read "...A. ervi females spent far less time in the central zone (“undetermined choice”) than in the branches containing odor stimuli (Fig. 2, Table 1).”
In lines 204 to 208, two sentences are not clear. I would say better “We obtained the same result when females were reared on the SaB HPC and switched for one generation on ApF HPC (Fig. 2c, Table 1).” And “On the other hand, females originating from the SaB HPC spent the same amount of time in the branches with odors of either HPC, whether they had been reared long-term on SaB HPC or reared on the ApF HPC and switched for one generation on the SaB HPC (Fig. 2b,d, Table 1).”
Also, in the discussion on the influence of environmental complexity I would also discuss on the “mother knows best” hypothesis i.e. parasitoid females should choose the host that will maximize fitness of progeny...
Author Response
The paper is well written and the study well conducted however I recommend some changes on the content of the manuscript. In fact, I suggest to completely delete the study on EAG.
Re: We thank the reviewer for his very helpful comments. We would, however, prefer to keep the EAG data in the manuscript, even if they provide only a minor advancement in knowledge. We have made changes in the text to clarify the EAG results and their interpretation and provide responses to the reviewer’s comments in more detail below. Should the reviewer insist on removing the EAG part of the study, we are, however, ready to do so.
The overall EAG study (and the results and discussion on this) is confusing in this study since it does not bring a strong inputs in the overall manuscript. The authors used commercial compounds but not compounds or blend coming directly from their HPC under their experimental and rearing conditions. In addition, concluding that “A. ervi females reared on the ApF HPC were less sensitive to volatiles associated to the SaB complex.” is abusive since they used commercial compounds but not from compounds coming directly from their HPC in their both experimental and rearing conditions.
Re: We agree that the EAG study only concerns a very small aspect as compared to the behavioural study, but we nevertheless think that our results contribute to our knowledge on sensory mechanisms of host plant complex differentiation. We chose individual compounds for practical reasons: dose-response relationships can be much more easily investigated for individual compounds than for complex blends or even more so for odours directly emanating from HPCs. As we were searching for differences in antennal sensitivity in order to know if individuals of different origin might exhibit differential sensitivity to odours emitted by different HPCs, the approach using individual components seemed appropriate in our eyes. We did not intend to claim that the differences in antennal sensitivity are sufficient to explain behavioural differences in attraction (see also below). And indeed the sentence you cite is formulated awkwardly, leading to wrongly over-interpreting our results. We reformulated the conclusions accordingly.
Moreover, concerning the EAG study, in Mat. & Met. section, the authors chose sulcatone and sulcatol for EAG done for the S. avenae/Barley complex based on the fact that these volatiles are emitted after aphid infestation in different cereal-aphid complexes BUT are they attractive to A. ervi as argued for the selection of the 2 volatiles used for the A.pisum/Faba bean complex? this aspect is not clear.
Re: We understand the criticism of the reviewer in the light of the way we presented the results. We certainly formulated the text parts concerning EAGs not clearly enough. We now reformulated all text parts on EAGs to clarify the message we would like to pass. In fact, a higher antennal sensitivity to compounds, which occur in one HPC or the other, does not necessarily imply that the behavioural response to these individual odours or even the blend emitted by the HPC needs to be stronger. A higher antennal sensitivity might cause as well a stronger attraction, as a stronger repellent effect. Therefore it is at the actual level of our investigation rather irrelevant if the individual compounds we tested in the EAG experiments are behaviourally attractive or not. Indeed our argumentation that volatiles chosen for the ApF HPC might be behaviourally attractive on their own is misleading, but we added now also information on potential attractiveness of sulcatone for the sake of completeness
We have now modified the sentence concerning EAG in the abstract (see also responses to reviewer 2) and in the last paragraph of the introduction, and re-written the material and methods, discussion and conclusions parts of the EAG work to clarify the points mentioned above and attenuate the importance, which we attributed to our results in the first version of the manuscript. We have also changed the title of the EAG discussion part to “The effect of experience of antennal sensitivity to HPC-emitted volatiles”.
In this context, I suggest to the authors to delete the entire EAG study of their manuscript (somehow this is a too preliminary study).
Re: As mentioned above, we would prefer to keep the EAG results in the manuscript, but we are ready to remove them if the reviewer insists.
I have also the minor points to address:
In the results section, in line 198 it might be Fig 2 and not only Fig. 2a? to read "...A. ervi females spent far less time in the central zone (“undetermined choice”) than in the branches containing odor stimuli (Fig. 2, Table 1).”
Re: We agree and changed the sentence accordingly (now lines 210-212).
In lines 204 to 208, two sentences are not clear. I would say better “We obtained the same result when females were reared on the SaB HPC and switched for one generation on ApF HPC (Fig. 2c, Table 1).” And “On the other hand, females originating from the SaB HPC spent the same amount of time in the branches with odors of either HPC, whether they had been reared long-term on SaB HPC or reared on the ApF HPC and switched for one generation on the SaB HPC (Fig. 2b,d, Table 1).”
Re: We changed these sentences accordingly (now lines 217-222).
Also, in the discussion on the influence of environmental complexity I would also discuss on the “mother knows best” hypothesis i.e. parasitoid females should choose the host that will maximize fitness of progeny...
Re: We agree that results on environmental complexity have interesting implications in the context of the “mother knows best” hypothesis. We added a few lines in the discussion accordingly (lines 359 to 367).
Reviewer 2 Report
The authors investigated the effect of two different host-plant complexes (HPCs) on which parasitoids were reared during their early life on their adult behavior. They also measured parasitoid antennal responses to four compounds specifically produced by one of the two HPCs. Finally, they tested the effect of the presence of one HPC (near the HPC used for rearing) during the early development of parasitoids on their adult choice.
They found that one HPC (ApF) significantly affects parasitoid adult behavior when they were reared on it. This was not the case for the other HPC (SaB). The antennal response only partially supported the behavioral results observed. Finally, they found a significant effect of the presence of an HPC on adult parasitoid behavior.
This manuscript is clear and well written. The subject is of great interest, the experiments seem well conducted and the results are very interesting. Please find below some detailed remarks.
Abstract
L22: “in accordance with behavioral differences observed”. This result is only partially correlated with the behavioral differences. A lower sensitivity of parasitoids (reared on Apf) to SaB volatiles may explain results from Fig. 2a, but do not explain results from Fig. 2b. I would be more cautious with this statement.
1) Introduction
L32: sentence too long
L60: consider using “Although” instead of “whereas”
2) Materials and Methods:
L114: please mention the origins
L129: This does not look like a valid justification for the choice of the central zone size. It should be based on what seems the more realistic in term of deciding whether a parasitoid position corresponds to a choice or not. In addition, it makes the sentence L196 quite obvious, as you specifically designed the center area to be small enough so parasitoids would not spend too much time in it compared to the other bigger areas.
L120: “Stimulation HPCs were changed every day”: so this means that the olfactometer tests were performed over several days. In L137, you mention the number of wasps used per group of parasitoid, but you do not mention if one group of parasitoid was tested in only one day, or during several days, which would include using different individual stimulation HPCs. Please clarify
L182: consider replacing “excluding this zone” by “excluding it”
3) Results
L223-224: delete “a significant effect of HPC origin occurred:”
L252-252: delete one sentence
4) Discussion
L305-306: If it was the case, you may expect A. ervi originating from the SaB to behaviorally respond more to the SaB HPC because they are more sensitive to compounds emitted by this complex, but this is not what you found.
Author Response
The authors investigated the effect of two different host-plant complexes (HPCs) on which parasitoids were reared during their early life on their adult behavior. They also measured parasitoid antennal responses to four compounds specifically produced by one of the two HPCs. Finally, they tested the effect of the presence of one HPC (near the HPC used for rearing) during the early development of parasitoids on their adult choice.
They found that one HPC (ApF) significantly affects parasitoid adult behavior when they were reared on it. This was not the case for the other HPC (SaB). The antennal response only partially supported the behavioral results observed. Finally, they found a significant effect of the presence of an HPC on adult parasitoid behavior.
This manuscript is clear and well written. The subject is of great interest, the experiments seem well conducted and the results are very interesting. Please find below some detailed remarks.
Re: We thank the reviewer for the very helpful comments on our manuscript.
Abstract
L22: “in accordance with behavioral differences observed”. This result is only partially correlated with the behavioral differences. A lower sensitivity of parasitoids (reared on Apf) to SaB volatiles may explain results from Fig. 2a, but do not explain results from Fig. 2b. I would be more cautious with this statement.
Re: We thank the reviewer for pointing out this point. We formulated this statement now more cautiously and it reads now: “Electroantennographic recordings revealed significant sensitivity differences for some of the tested individual volatiles, which are emitted differentially by the two HPCs.”
1) Introduction
L32: sentence too long
Re: We separated this sentence in two now: “Whereas specialist species generally rely on innate preferences [3], such innate preferences for the most suitable host in generalist species are in many cases modified by experience. The influence of experience allows thus to take host availability into account [4,5].”
L60: consider using “Although” instead of “whereas”
Re: done
2) Materials and Methods:
L114: please mention the origins
Re: We now added the two origins and the three types of rearing conditions in the sentence.
L129: This does not look like a valid justification for the choice of the central zone size. It should be based on what seems the more realistic in term of deciding whether a parasitoid position corresponds to a choice or not. In addition, it makes the sentence L196 quite obvious, as you specifically designed the center area to be small enough so parasitoids would not spend too much time in it compared to the other bigger areas.
Re: We reformulated this sentence accordingly: “This central zone was designed as narrow as possible, but allowing a realistic decision on whether a parasitoid position corresponds to a choice or not.”
L120: “Stimulation HPCs were changed every day”: so this means that the olfactometer tests were performed over several days. In L137, you mention the number of wasps used per group of parasitoid, but you do not mention if one group of parasitoid was tested in only one day, or during several days, which would include using different individual stimulation HPCs. Please clarify
Re: We added the information that each group of parasitoids had been tested on at least 3 experimental days.
L182: consider replacing “excluding this zone” by “excluding it”
Re: done
3) Results
L223-224: delete “a significant effect of HPC origin occurred:”
Re: done
L252-252: delete one sentence
Re: done
4) Discussion
L305-306: If it was the case, you may expect A. ervi originating from the SaB to behaviorally respond more to the SaB HPC because they are more sensitive to compounds emitted by this complex, but this is not what you found.
Re: You are absolutely right. We deleted this sentence now. See also responses to reviewer 1.
Reviewer 3 Report
My annotations to manuscript are attached. I provided numerous editorial changes, mostly minor. There are a couple of comments authors need to address. Again, these have mainly to do with clarity of presentation.

Author Response
My annotations to manuscript are attached. I provided numerous editorial changes, mostly minor. There are a couple of comments authors need to address. Again, these have mainly to do with clarity of presentation.
Re: We thank the reviewer for his/her very helpful comments. We made all editorial changes accordingly. Regarding the more general comments, we made the following modifications:
L. 19-21. We changed ”We revealed an asymmetric olfactory host fidelity when females experienced a simple environment” to “In a simple environment, HPC of origin had an influence on olfactory choice, but the preferences observed were asymmetric according to parasitoid origin.”
L. 57-60: We changed the sentence to: “This induction of host preference coincides with transcriptomic differences between females reared on different HPCs: genes involved in neuronal growth and development, signalling pathways and olfactory detection are differentially expressed according to the HPC of origin [17].”
L. 99: We think there may have been a printing problem here. The sentence was “That way, females used for experiments were exposed only to the odor cues of their natal HPC during their larval and pupal development.”
We changed the sentence nevertheless to: “Thus females used for experiments…”
L. 211-213: This first sentence on behavioural results summarizes the general findings for insects of both origins. The sentence suggested by referee 3 is thus not correct as it stands (females spend the same amount of time or less time in the SaB branches, depending on the origin). We changed the sentence thus only partly to: “Independent of their conditioning, A. ervi females never spent significantly more time in the branches with odors from the SaB HPC than in the branches with odors from the ApF HPC (Fig 2).”
L. 281: We prefer to keep the sentence as it is: “genetic” Is already mentioned earlier in the same sentence.
Round 2
Reviewer 1 Report
The authors responded satisfactorily to my comments. I am then OK to keep the EAG results in the manuscript as they suggested.